# Tea Polyphenols Protects Tracheal Epithelial Tight Junctions in Lung during *Actinobacillus pleuropneumoniae* Infection via Suppressing TLR-4/MAPK/PKC-MLCK Signaling

**DOI:** 10.3390/ijms241411842

**Published:** 2023-07-24

**Authors:** Xiaoyue Li, Zewen Liu, Ting Gao, Wei Liu, Keli Yang, Rui Guo, Chang Li, Yongxiang Tian, Ningning Wang, Danna Zhou, Weicheng Bei, Fangyan Yuan

**Affiliations:** 1National Key Laboratory of Agricultural Microbiology, College of Veterinary Medicine, Huazhong Agricultural University, Wuhan 430070, China; lixiaoyue1208@webmail.hzau.edu.cn (X.L.); aningaha0802@163.com (N.W.); 2Cooperative Innovation Center of Sustainable Pig Production, Wuhan 430070, China; 3Hubei Hongshan Laboratory, Wuhan 430070, China; 4Key Laboratory of Prevention and Control Agents for Animal Bacteriosis (Ministry of Agriculture and Rural Affairs), Hubei Provincial Key Laboratory of Animal Pathogenic Microbiology, Institute of Animal Husbandry and Veterinary, Hubei Academy of Agricultural Sciences, Wuhan 430064, China; liuzwen2004@sina.com (Z.L.); gaotingyefeiyeziyu@163.com (T.G.); liuwei85@126.com (W.L.); keliy6@126.com (K.Y.); hlguorui@163.com (R.G.); lichang1113@hbaas.com (C.L.); tyxanbit@163.com (Y.T.); zdn_66@126.com (D.Z.)

**Keywords:** tea polyphenols, *Actinobacillus pleuropneumoniae*, epithelial barrier, TLR-4/MAPK/PKC-MLCK signaling

## Abstract

*Actinobacillus pleuropneumoniae* (APP) is the causative pathogen of porcine pleuropneumonia, a highly contagious respiratory disease in the pig industry. The increasingly severe antimicrobial resistance in APP urgently requires novel antibacterial alternatives for the treatment of APP infection. In this study, we investigated the effect of tea polyphenols (TP) against APP. MIC and MBC of TP showed significant inhibitory effects on bacteria growth and caused cellular damage to APP. Furthermore, TP decreased adherent activity of APP to the newborn pig tracheal epithelial cells (NPTr) and the destruction of the tight adherence junction proteins β-catenin and occludin. Moreover, TP improved the survival rate of APP infected mice but also attenuated the release of the inflammation-related cytokines IL-6, IL-8, and TNF-α. TP inhibited activation of the TLR/MAPK/PKC-MLCK signaling for down-regulated TLR-2, TLR4, p-JNK, p-p38, p-PKC-α, and MLCK in cells triggered by APP. Collectively, our data suggest that TP represents a promising therapeutic agent in the treatment of APP infection.

## 1. Introduction

Tea is a popular beverage consumed worldwide. It is made from the leaves of the plant *Camellia sinensis*, which originated in ancient China [1,2]. Tea polyphenols (TP) are a specific bioactive ingredient of tea, and have various health promoting properties [3], including antioxidant, antimutagenic, immuno-regulatory, hypocholesterolemic, antibacterial, and anticancer activities [4,5]. If TP reach sufficient concentrations after drinking tea for a long time, they are absorbed and retained and exert their desired effects in plasma and tissues [1]. Supplementation with TP mainly alters the gut microbiome composition and can benefit bone health [6], which can control obesity and related metabolic disorders [3,7]. TP have protective effects on high glucose-induced cell proliferation and senescence in human glomerular mesangial cells (HGMCs) [8]. In addition, several studies have shown that natural polyphenols not only have antibacterial effects, but also show low toxicity and great bioavailability [9,10]. TP show great promise as antibiotic alternatives with good antibacterial effects. Previous studies have shown that TP can lower the secretion of pro-inflammatory cytokines and reduce the inflammatory response to *Fusobacterium nucleatum* [11], inhibit the virulence of *Pseudomonas aeruginosa* [12], and protect against *Haemophilus parasuis* challenge [13]. However, the effects of TP on *Actinobacillus pleuropneumoniae* infection are not completely understood.

*Actinobacillus pleuropneumoniae* (APP) is one of the most common bacterial pathogens causing porcine respiratory infections. It is an etiological agent for porcine pleuropneumonia, which is characterized by acute hemorrhagic, purulent, and fibrous pleuropneumonia symptoms [14]. APP can infect pigs of all ages, colonize the upper respiratory tract, and breach the epithelial barrier to cause local or systemic infection [15]. The morbidity of the resulting disease can be as high as 100%, but generally varies between 30–50% [16], and causes considerable economic losses in the swine rearing industry [17]. It comprises two biotypes based on their dependence on nicotinamide adenine dinucleotide (NAD) (biotype 1, biotype 2) [18]. At present, 19 different serotypes of APP have been recognized based on their polysaccharide compositions [19], with serovars 1, 5, 9, and 11 considered the most virulent [20]. Serotypes 1, 3, 4, 5, and 7 are typically isolated in China [21]. The varieties and prevalence of the serotypes vary between most regions of China. Owing to the diversity of serotypes and differences in their regional prevalence, there is currently no satisfactory vaccine to control outbreaks of APP infection and antibiotics remain the most effective means of control in most regions [22]. For pigs, APP infections are often treated with macrolides, β-lactams, fluoroquinolones, and/or florfenicol in the swine industry [23,24,25]. From 2002 to 2013, a total of 71 APP isolates from pig farms in Australia showed a high frequency of resistance to erythromycin and tetracycline [26]. Among 162 APP strains collected from pigs in Spain from 2017 to 2019, a highly antibiotic resistance to doxycycline was discovered [23]. However, some APP strains have begun to show varying degrees of antibiotic resistance, and this presents problems in controlling outbreaks of porcine pleuropneumonia [27,28,29], leading to an urgent need for alternatives to antibiotics.

Infection with APP can damage the porcine respiratory epithelial barrier [30]. Tracheal epithelial cells play a defensive role in the epithelial barrier [31]. The epithelium is composed of adherent cells, which are polarized and have an apical domain and a basolateral domain [32]. It forms a physical barrier between the internal and external environment, protecting against environmental contaminants and pathogens [33,34]. The epithelial cells are tightly joined by a set of intercellular junctions composed of gap junctions, desmosomes, tight junctions (TJs), and adherence junctions (AJs) [32,35]. Epithelial tight junctions are located at the apicolateral boundary of the epithelial cells and form the paracellular barrier. This barrier regulates epithelial permeability and the intramembrane barrier, which separates the membrane components [32,36]. They are composed of at least three membrane proteins, including zonula occludens-1 (ZO-1), claudins, and occludin [37,38]. Claudins and occludin are polytopic membrane proteins with four transmembrane domains. The consensus is that claudins mainly modify the pore pathways, and that ZO-1, occludin, and tricellulin regulate the leak pathways [36,39]. Adherence junctions are located directly beneath the TJs and provide intercellular adhesion to maintain epithelial integrity [40,41]. They are composed of E-cadherin, nectin, and α and β-catenin, which have diverse functions, including the maintenance of actin binding, cells polarization, and signal transduction-related transcriptional regulation [42,43,44]. The formation, dismantling, and maintenance of TJs are regulated, in part, by phosphorylation and dephosphorylation of TJ proteins as well as some signaling pathways, including protein kinase C (PKC), myosin light chain kinase (MLCK), and mitogen-activated protein kinases (MAPK) [45,46,47,48].

Pleuropneumonia infection causes substantial economic losses to the swine rearing industry worldwide, and its various strains exhibit varying degrees of antibiotic resistance, causing an urgent need for alternatives to antibiotics in combatting APP infection [27,28,29]. Despite the aforementioned antimicrobial potential of TP, their effectiveness against APP infection remains to be elucidated. Other studies report that infection with APP can damage the porcine respiratory epithelial barrier [30], although data regarding the effects of TP on the respiratory epithelium remain limited. This study investigated the possible mechanisms by which the application of TP might affect the growth and virulence of APP and its effect on the epithelial barrier in order to provide a new strategy for preventing APP infection in pigs.

## 2. Results

### 2.1. TP Inhibit the Growth of APP In Vitro

The results of the micro-broth dilution assay showed that TP exhibited antibacterial activity against APP (Table 1). The MIC and MBC values of TP were 0.625 mg/mL and 1.25 mg/mL, respectively. The growth properties of APP strains and TP (0 MIC, 0.5 MIC, 1 MIC, and 2 MIC) when co-cultured for 0, 1, 2, 3, 4, and 5 h were investigated. Cultures were serially diluted with PBS and plated onto TSA, and the number of bacteria were counted. The growth curves of the APP strains showed that TP exhibited a dose-dependent bactericidal effect on APP. The growth of APP was significantly inhibited after co-culture with TP (1 MIC) for 5 h (*p* < 0.001) (Figure 1A). The MIC and MBC values of florfenicol were 1 μg/mL and 2 μg/mL, respectively. The addition of florfenicol to bacteria was used as a control (Figure 1A). 

### 2.2. TP Affect the Cellular Integrity of APP 

The effect of TP on the cell integrity of APP was observed using a TEM and an SEM. APP cells treated with TP (0.625 mg/mL) showed obvious damage when compared with the control cells (Figure 1B). The treated cells showed instances of cell wall damage and ruptured membranes, accompanied by the leakage of cytoplasmic contents.

### 2.3. TP Effect on the Adhesion of APP Bacteria 

Figure 1C shows the results of LDH release from NPTr cells treated with different concentrations of TP. Different concentrations of TP had little effect on the toxicity of NPTr cells. As shown in Figure 1D, pretreatment with TP affected the capacity of APP strains to adhere to NPTr cells. TP inhibited the ability of APP to adhere to NPTr cells.

### 2.4. Effect of TP on Pro-Inflammatory Cytokine Secretion and the mRNA Level of TLR2 and TLR4 in NPTr Cells 

TP influenced the secretion of pro-inflammatory factors in NPTr cells infected by APP. As shown in Figure 2A–C, secretion of IL-6, IL-8, and TNF-α pro-inflammatory factors in NPTr cells infected by APP was significantly higher than in the control group. However, pretreatment with TP for 3 h reduced the secretion of IL-6, IL-8, and TNF-α inflammatory factors in NPTr cells.

TP influenced the mRNA level of TLR2 and TLR4 in NPTr cells infected by APP. As shown in Figure 2D,E, the secretion of TLR2 and TLR4 in NPTr cells stimulated by APP was significantly than in the control groups. However, the secretion of TLR2 and TLR4 in cells pretreated with TP was significantly lower than in cells treated with APP alone (Figure 2D,E). These results suggest that TP could inhibit the secretion of TLR2 and TLR4 in NPTr cells infected by APP.

### 2.5. TP Decrease the Disruption of Cellular Junctions in NPTr Cells 

The effect of TP on the integrity of the epithelial barriers infected by APP was further evidenced by the β-catenin and occluding levels of the two important TJ and AJ proteins. The localization of immunofluorescence of ß-catenin and occluding was observed by microscope. As shown in Figure 3, there was a destructive effect on the NPTr cell TJs infected by APP when compared with the control cells, and β-catenin and occludin expression was downregulated. However, when NPTr cells were pretreated with TP, the downregulation of these expressions was inhibited. This result was confirmed by immunoblotting. As shown in Figure 4B, protein expression levels of β-catenin and occludin were significantly decreased post-infection by APP in contrast to those of uninfected control cells. Following pretreatment of NPTr cells with TP, the protein expression levels of β-catenin and occludin were higher than those in untreated cells stimulated by APP. These results showed that pretreatment of cells with TP can inhibit the destruction of the β-catenin and occludin proteins, respectively, by APP.

### 2.6. The Effect of TP on the Expression of Toll-like Receptor-Related Proteins in NPTr Cells

TP influenced the proteins expression level of TLR2 and TLR4 in NPTr cells infected by APP. As shown in Figure 4A, the expression level of TLR2 protein in NPTr cells stimulated by APP was significantly lower than that in the control groups. However, the protein expression levels of TLR2 in cells pretreated with TP was significantly lower than in cells treated with APP alone. For the pretreatment of NPTr cells with TP, the protein expression levels of TLR4 were significantly lower than those in treated cells stimulated by APP. These results suggest that TP could inhibit the upregulation of TLR2 and TLR4 protein expression in NPTr cells infected by APP.

### 2.7. Effects of TP on PKC-α and MLCK Signaling Pathway Activated by APP

The protection mechanisms of TP on TJ of APP infected NPTr cells were determined by measuring related protein expressions using Western blot assays. The phosphorylation level of PKC-α was significantly increased under APP infection (*p* < 0.001). TP could downregulate the expression of p-PKC-α induced by APP (Figure 4C). The MLCK protein was significantly increased in the NPTrs in APP group when compared to control group (*p* < 0.0001). TP could inhibit the expression of MLCK protein (Figure 4D).

### 2.8. Effect of TP on the MAPK Signaling Pathway Activated by APP

We investigated the effect of APP on activation of the MAPK signaling pathway in NPTr cells using Western blot analysis. The NPTr cells were infected with APP, and phosphorylation of MAPK was measured using phospho-specific Abs. The data showed that APP promoted the phosphorylation of JNK and p38 when compared with that in the control condition (Figure 4E), whereas the phosphorylation of JNK and p38 in NPTr cells challenged with APP was inhibited by TP (Figure 4F).

### 2.9. Protective Effect of TP on Mice Infected by APP

In a mouse model, the protective effect of TP was assessed by instillation of TP prior to infection with APP. The results showed that the mice treated with TP suffered reduced inflammation and had higher survival rates when compared with untreated mice infected with APP. TP were demonstrated to confer protective effects against a lethal dose of APP (Figure 5A). We analyzed pathological samples of lung tissues of mice in the different groups and collected serum to detect pro-inflammatory factors. The levels of serum pro-inflammatory factors in the mice showed that the secretion of IL-1β, IL-6, and TNF-α inflammatory factors significantly increased in mice infected APP. Pretreatment with TP reduced the secretion of IL-1β, IL-6, IL-8, and TNF-α inflammatory factors, though the differences do not seem to be significant (Figure 5B). As shown in Figure 5C, the tissues of mice infected with APP showed abnormal lung tissue structure, partial alveolar atrophy, alveolar wall thickening, and some protein fluid and inflammatory cell infiltration when compared with the control group. However, symptoms were reduced in the lung tissues of mice treated with TP.

## 3. Discussion

Porcine pleuropneumonia is a common respiratory disease caused by APP infection. There are 19 different serotypes of APP, and the current vaccines does not cover all the serotypes. Antibiotic treatment remains an effective measure, but some strains have begun to show varying degrees of antibiotic resistance. Antibiotic resistance is a major global problem and there is an urgent need to develop new therapeutics. Considerable interest has been shown in the potential of botanical medicines to prevent and alleviate diseases, and these show great promise as antibiotic alternatives.

Tea is one of the most frequently consumed beverages in the world and has a long and rich history of medicinal benefits [49]. While there are multiple factors of tea influencing the effective biological properties, tea polyphenols are the most significant and valuable components [50]. Previous research has shown that TP have strong antibacterial properties and show significant promise as antibacterial agents in combating bacterial diseases [10,13,51]. TP is a kind of pure natural biological active substance extracted from green tea; the content is 98% in certificate of analysis. TP consist of different sorts of compounds. There are four major tea catechins, including epigallocatechin gallate (EGCG, 31.12%), epicatechin gallate (ECG, 20.31%), epicatechin (EC, 7.83%), and epigallocatechin (EGC, 6.02%), in accord with compounds about tea catechins shown in previous studies [50]. This provides some theoretical basis for achieving the observed biological effect. In this study, TP extracted from green tea exhibits antimicrobial activity towards APP. The determination of MIC, MBC, and the growth curves of APP strains showed that TP had a dose-dependent bactericidal effect on APP. The influence of TP on the integrity of APP cells was observed using a TEM, which confirmed that APP cells treated with TP showed obvious damage, in accord with those of previous studies [10,13,51,52]. In this study, we used a mouse model to examine the influence of APP on lung tissues. APP associated lung damage was reduced, and the survival rate was higher in mice fed TP when compared with untreated mice. The levels of serum pro-inflammatory factors detected in mice showed that pretreatment with TP reduced the secretion of IL-6, IL-8, and TNF-α inflammatory factors (Figure 5B). These results suggest that TP confer protection against APP infection and may provide a new means of disease prevention and treatment. Previous studies have used NPTr as a cellular model to study the role of pathogens in porcine respiratory diseases [53,54,55]. In this study, pretreatment of NPTr cells with TP showed that they inhibited the ability of APP to adhere to NPTr cells, in concurrence with previous studies [11,52,56].

Tracheal epithelial cells play a significant role in airway defense under multiple pathogen attacks [31]. TJs regulate the passage of ions and molecules through paracellular pathways in epithelial and endothelial cells [47]. TJ and AJ proteins have multiple functions and play important roles in maintaining epithelial barrier integrity [32,36,40,41]. However, there are little data available on how TJ and AJ changes under APP infection and their interaction mechanism. To explore the effect of TP on the integrity of the epithelial barrier against APP infection, we examined the levels of two important TJ and AJ proteins, β-catenin and occludin. Immunofluorescence analysis showed that APP infection altered the localization of β-catenin and occludin in epithelial cells and disrupted the TJ and AJ of NPTr cells. The distributions of occludin and β-catenin protein of APP infected NPTr cells were significantly disrupted. These data provide evidence that APP infection can alter the AJ and TJ and damage the epithelial barrier. The damage to junction proteins was alleviated when we pretreated the cells with TP, showing that TP can prevent the abnormalities caused by APP. This is in accordance with the results of previous studies that show that TP promote the expression of connexins, thereby protecting them from pathogens, a process largely prevented by TP supplementation [52,57,58,59].

It has been shown that PKCs can regulate the epithelial and endothelial barriers through their regulatory effects as intracellular signaling molecules [48]. Activation of PKC will increase cell permeability, which plays an important role in the regulation of tight junctions [48]. This study confirmed the changes of PKC-α protein in NPTr cells infected with APP. Compared with the control group, the expression of phosphorylation of PKC-α of APP-infected was significantly increased. TP can attenuate the phosphorylation of PKC-α. The results suggest that the protection effect of TP of tight junction abnormalities may relate directly to inhibition of PKC and/or the downstream signaling pathways such as MLCK.

Myosin light chain kinase (MLCK) is a key signaling node in physiological and pathophysiological regulation of epithelial tight junctions [60]. MLCK has been demonstrated to be the most important factor that influence TJ during inflammation. Increased MLCK is an indicator of TJ barrier disruption and can be triggered by pro-inflammatory cytokines, including TNF-α, IL-1β, and several related molecules [60]. Our results showed that the MLCK protein was significantly increased in APP challenged NPTr cells, suggesting that APP infection could cause TJ barrier disruption. However, TP inhibited the protein levels of MLCK induced by APP. The results indicate that the protective effects of TP on TJ may derive from inhibiting the MLCK pathway. In order to comprehend the protective effect of TP on epithelial barrier, further studies must be conducted.

An increasing number of cytokines have recently been confirmed to influence TJ barrier function, and to be associated with intrinsic TJ proteins [61]. Previous research has shown that IL-1β is the hallmark innate cytokine that plays a key role in the initiation of inflammatory immune responses and has been associated with inflammatory cell migration and osteoclastogenesis [62,63]. IL-6 is rapidly and transiently produced in response to infections and tissue injuries, and is implicated in inflammation, hematopoiesis, and immune responses [64]. IL-8 is a chemokine mainly produced by monocytes and epithelial cells, and mediates the chemotaxis of neutrophils during acute phase of inflammation [65]. TNF-α is a multi-function cytokine and an important mediator of inflammatory responses, produced by many types of immune cells including mucosal cells (such as epithelial cells) [66]. It regulates a number of inflammatory signaling pathways in macrophages [67]. Previous studies have demonstrated that APP can induce breakdown of the integrity of the porcine tracheal epithelial barrier, allowing tracheal epithelial cells to secrete various cytokines [30]. In this study, porcine tracheal epithelial cells also produced IL-6, IL-8, and TNF-α following exposure to APP, a result consistent with previous studies [30,53]. However, pretreatment with TP decreased the secretion of inflammatory factors IL-6, IL-8, and TNF-α in porcine tracheal epithelial cells.

Activation of the MAPK pathway can lead to TJ and AJ opening or assembly. To our knowledge, this is the first report showing TP inhibiting inflammatory responses inhibiting p38 MAPK signaling and the TLR signaling pathway in tracheal epithelial cells (Figure 6). JNK play an important role in regulating cell viability [68]. It has been reported that activation of inflammatory signaling pathways, including JNK and p38, induces secretion of cytokines [69]. In our study, TP decreased the secretion of inflammatory factors IL-6, IL-8, and TNF-α. APP activated the inflammatory signaling molecules JNK and p38 in NPTr cells. TP treatment significantly reduced APP-induced JNK and p38 phosphorylation expression. It is suggested that TP potentially mediates activation of the p-p38 MAPK pathway induced by APP. These data suggest that TP modulate the inflammatory immune response to fight infection and improve healing.

Toll-like receptors (TLRs) are transmembrane pattern recognition receptors (PRRs) that play a key role in microbial recognition, systemic bacterial infection, and control of adaptive immune responses [70,71]. Toll-like receptor 2 (TLR2), one member of the TLR family, recognizes conserved molecular patterns related to both Gram-negative and Gram-positive bacteria, such as lipoteichoic acid (LTA), lipoarabinomannan, lipoproteins, and peptidoglycan (PGN) [72,73]. Toll-like receptor 4 (TLR4) plays a crucial role in the infective inflammation caused by Gram-negative bacteria [74,75]. Related research shows that Toll-like receptor 4 plays a key role in mediating the innate immune response to pneumonia infection [75,76]. A previous study demonstrated that Emodin inhibits influenza viral pneumonia by inhibiting IAV-induced activation of TLR4, MAPK, and NF-kB pathways [77]. Ugonin M might exert efficacy on LPS-induced lung infection and inhibit not only NF-kB and MAPK activation but also TLR4 protein expression [78]. In the present study, mRNA levels of TLR2 and TLR4 in NPTr cells infected with APP were higher than those in control cells. When NPTr cells were pretreated with TP, a similar result was obtained and the TLR2 and TLR4 mRNA level of NPTr cells infected with APP were suppressed and the protein expression levels of TLR2 and TLR4 were significantly lower than those in treated cells stimulated by APP.

## 4. Materials and Methods

### 4.1. Cell Culture, Bacterial Culture, and TP

Newborn pig tracheal epithelial cells (NPTr) were cultured in Dulbecco’s modified eagle medium (DMEM, high-glucose: Cytiva, Washington, DC, USA) supplemented with 10% fetal bovine serum (Gibco, New York, NY, USA), 100 U/mL penicillin, and 100 U/mL streptomycin (Solarbio, Beijing, China) at 37 °C with 5% CO_2_. APP serotype 5b was obtained from the State Key Laboratory of Agricultural Microbiology (Huazhong Agricultural University, Wuhan, China). The bacterial strains were cultivated in BD™ tryptic soy broth (TSB) and tryptic soy agar (TSA) or Mueller–Hinton (Beckton Dickson, New York, NY, USA) supplemented with sterile newborn calf serum (10%, *v*/*v*, AusGeneX, Brisbane, Australia) and 10 µg/mL nicotinamide adenine dinucleotide (NAD) at 37 °C. TP (purity: 98%, molecular weight: 281.36) were bought from the Nanjing Tianrun Biotechnology Co., Ltd. (Nanjing, China). This product is a kind of pure natural biological active substance extracted from tea; its main components are epigallocatechin gallate (EGCG), epigallocatechin (EGC), epicatechin gallate (ECG), and epicatechin (EC).

Determination of minimum inhibitory concentration (MIC) and minimum bactericidal concentration (MBC).

The determination of MIC and MBC values of TP against APP were tested using a modified broth micro-dilution assay as described by the Clinical and Laboratory Standards Institute, CLSI 2015 [12]. Serial two-fold dilutions of TP (from 160 mg/mL) in culture medium were placed in 96-well micro-titer plates, 100 μL per well. Each well was seeded with APP at a final concentration of 5 × 10^5^ CFU/mL. Medium samples without bacteria or TP were placed in wells as controls. The MIC value was the lowest concentration of TP at which no bacterial growth was observed after 12 h at 37 °C. Samples of 10 μL aliquots per well were spread onto TSA plates and left for 24 h at 37 °C, and the lowest concentration in which no APP colony formed was taken as the tea polyphenol MBC.

### 4.2. TP and APP Co-Cultivation Affects Bacteria Growth

The effect of TP on APP growth was explored as previously described [5], with minor modifications. In summary, APP (1 × 10^8^ CFU/mL) and TP (0 MIC, 1/2 MIC, 1 MIC, and 2 MIC) were co-cultured for 0, 1, 2, 3, 4, and 5 h. At each hourly point, cultures were diluted, seeded onto TSA plates, and incubated for 24 h at 37 °C. The bactericidal effects of TP against APP were defined by measuring the CFUs of each culture by counting the number of APP colonies. The log10 CFU/mL vs. time over a 5 h period was plotted to visualize the APP growth curves.

Transmission electron microscope (TEM) and Scanning electron microscope (SEM) analysis of APP cellular integrity.

The damage caused to APP cells by TP was detected using TEM and SEM as previously described [79], with modifications. APP was grown in TSB until OD_600_ to 0.6, then bacteria were incubated with TP (0.625 mg/mL) at 37 °C for 3 h, and then fixed with electron microscope fixative (2.5% glutaraldehyde) at 4 °C overnight. The bacteria samples were post-fixed in 1% osmic acid for 2 h at room temperature, and ethanol and acetone were then added in turn for dehydration. Finally, ultra-thin sections were embedded and dyed with uranium and lead double staining. The morphology of the cells was observed under a TEM (Hitachi, HT7700, Tokyo, Japan), or the dried samples were observed via a SEM (Hitachi, SU8010, Tokyo, Japan); images were collected for analysis.

### 4.3. Cytotoxicity Detection Assay

In the cytotoxicity detection assay, NPTr cells were seeded into 96-well plates, with 10^6^ cells per well. Different concentrations of TP were added to the cells as treatment groups, and cells without added TP were used as negative controls. The viability of the NPTr cells under different conditions were detected by measuring the amount of lactose dehydrogenase (LDH) using an LDH assay kit (Beyotime, Shanghai, China) according to the manufacturer’s protocols. The LDH viability in the supernatant was measured using a microplate reader (Molecular Devices, Sunnyvale, CA, USA) at 490 nm.

### 4.4. Adherence Assay

In the adherence assay, NPTr cells were seeded into 24-well culture plates, with 10^6^ cells per well. After incubation overnight, control cells were not pretreated, and treatment group cells were pretreated with TP (0.0625 mg/mL) for 3 h. Cells were then infected with APP at a multiplicity of infection (MOI; bacterial cells per cell) of 10:1. Plates were incubated for 2 h at 37 °C in 5% CO_2_ and washed three times with PBS to remove unadhered bacteria. The cells with adherence bacteria were lysed with 0.025% Triton X-100 (Sinopharm Chemical Reagent Co., Ltd., Shanghai, China) on ice for 15 min. The number of bacteria adhering to NPTr cells were calculated.

### 4.5. Analysis of Cytokine, TLR2 and TLR4 mRNA Expression Using qRT-PCR

For analysis, 10^6^ NPTr cells per well were seeded into 12-well plates. NPTr cells were treated and infected as described above, with untreated cells serving as controls. One group of cells were pretreated with TP (0.0625 mg/mL) for 3 h, and the other groups were not pretreated. Cells were then infected with APP at a multiplicity of infection (MOI; bacterial cells per cell) of 10:1. Plates were incubated at 37 °C in 5% CO_2_. After 2 h, the total RNA of NPTr cells was extracted using Trizol reagent (Invitrogen, Burlington, ON, Canada) according to the manufacturer’s protocols. The cDNA was amplified using reverse transcriptase (Vazyme, Nanjing, China) and qRT-PCR with a SYBY Green qPCR Kit (Vazyme, Nanjing, China) and were performed in triplicate. Expression of every gene was normalized to glyceraldehyde 3-phosphate dehydrogenase (GAPDH). The sequences of primers used for the qRT-PCR analyses are listed in Table 2.

### 4.6. Immunofluorescence Assay

NPTr cells were treated and infected as described above. Untreated cells were used as controls. One group of cells were pretreated with TP (0.0625 mg/mL) for 3 h; the other group was not pretreated. Cells were then infected with APP at a multiplicity of infection (MOI; bacterial cells per cell) of 10:1. Plates were incubated for 2 h at 37 °C in 5% CO_2_. The cells were fixed in 4% paraformaldehyde and blocked in 5% BSA in PBS-Tween 20 (PBS containing 0.1% Tween 20) for 2 h at 37 °C. The cells were then labelled with antibodies against β-catenin (Proteintech, Chicago, IL, USA) and occludin (Proteintech, Chicago, IL, USA) at 4 °C overnight. After washing, cells were treated with a secondary antibody (CyTM3 AffiniPure Goat Anti-Mouse IgG (H + L)) (Jackson, PA, USA) and incubated for 1 h at room temperature. They were then washed with PBS three times, and the cell nuclei were counterstained with DAPI staining solution. The slides were then sealed with a coverslip using nail polish and kept in the dark until used. TJ and AJ proteins were visualized using a Nikon Eclipse CI fluorescence microscope and Nikon DS-U3 imaging (Nikon, Tokyo, Japan).

### 4.7. Western Blotting

NPTr cells were treated and infected as described above. Untreated cells were used as controls; one group of cells were pretreated with TP (0.0625 mg/mL) for 3 h and the other groups were not pretreated. Cells were then infected with APP at a multiplicity of infection (MOI; bacterial cells per cell) of 10:1. Plates were incubated at 37 °C for 2 h in 5% CO_2_. After 2 h, cells were lysed with RIPA lysis buffer (Beyotime, Shanghai, China) with added protease inhibitors. The protein concentrations were measured with a BCA protein assay kit (Beyotime, Shanghai, China). After SDS-PAGE separation, the protein samples were transferred to PVDF membranes and blocked in Tris-buffered saline/Tween 20 (TBST) containing 5% skim milk (Beckton Dickson, New York, NY, USA). The PVDF membranes were incubated overnight with the corresponding antibodies (Occludin Monoclonal antibody, Beta Catenin Monoclonal antibody, HRP-conjugated βeta actin antibody, TLR2 Monoclonal antibody, TLR4 Monoclonal antibody, MLCK Polyclonal antibody, JNK Monoclonal antibody, Phospho-JNK (Tyr185) Recombinant antibody, p38 MAPK Monoclonal antibody, and Phospho-p38 MAPK (Thr180/Tyr182) Polyclonal antibody, Proteintech, Chicago, IL, USA; PKC alpha Antibody, Phospho-PKC alpha (Ser657) Antibody, Affinity Biosciences, Cincinnati, OH, USA) at 4 °C. Then, the PVDF membranes were washed with TBST and incubated with HRP-conjugated secondary antibodies (HRP-conjugated Affinipure Goat Anti-Mouse IgG (H + L), HRP-conjugated Affinipure Goat Anti-Rabbit IgG (H + L), Proteintech, Chicago, IL, USA) for 1 h at room temperature and visualized with ECL solution (Vazyme, Nanjing, China). Finally, the PVDF membranes were observed using the ChemoDoc™ Touch Imaging System (Bio-Rad, Watford, UK).

### 4.8. Animal Assay

A total of 18 female BALB/c mice (6-weeks-old) were purchased from the Center for Disease Control of Hubei Province (Hubei CDC, Wuhan, China). All animal experiments followed the recommendations of the Laboratory Animal Monitoring Committee of Huazhong Agricultural University. The mice were randomly divided into three groups (six per group). One group was treated with TP (100 mg/kg) by oral gavage for 5 days. The two other groups were treated with equivalent distilled water. The group pretreated with TP and one of distilled water group were infected with APP (1.46 × 10^8^ CFU) by intraperitoneal injection. The other group were treated with normal saline by intraperitoneal injection, as a control group. Blood and lungs were obtained from the mice. Lung tissues were fixed in 4% para-formaldehyde and used for histopathological analysis. The amounts of inflammatory factors, including IL-1ß, IL-6, IL-8, and TNF-α, in the serum of the mice were determined using an ELISA Kit (ml063132-J, ml063159, ml063162, ml063162, mlbio, Shanghai, China) according to the manufacturer’s protocols.

### 4.9. Statistical Analysis

The results were analyzed using various statistical tests in GraphPad Prism version 8 (GraphPad Software, San Diego, CA, USA). Student’s *t*-test was used to analyze differences between groups; ‘*’ indicates statistical significance at *p* < 0.05, ‘**’ indicates statistical significance at *p* < 0.01, ‘***’ indicates statistical significance at *p* < 0.001, and ‘****’ indicates statistical significance at *p* < 0.0001.

## 5. Conclusions

In summary, our results demonstrated that TP inhibited the growth of APP, disrupting the integrity of APP cells. In a mouse model, TP reduced damage due to APP and gave protection against APP exposure. In addition, it was found that TP inhibited the ability of APP to adhere to NPTr cells. TP promoted the expression of junction proteins to conserve the epithelial barrier integrity. It is possible that the protective effects of TP on TJ are closely related to the inhibition of the activation of PKC and MLCK pathways. Our results suggested that APP induces MAPK signaling pathway activation and TP inhibits the MAPK signaling pathway via regulation of the inflammatory immune response. Our results suggest that TP were demonstrated to confer protective effects against a lethal dose of APP, and that they can be used as a replacement for antibiotics to provide a new strategy for preventing APP infection in pigs.

## Figures and Tables

**Figure 1 ijms-24-11842-f001:**
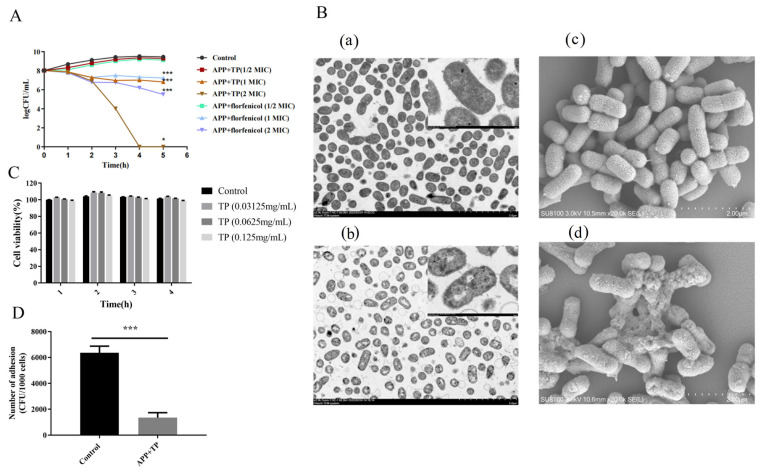
(**A**): Kinetics of the antimicrobial effects of TP on APP. APP and TP (0 MIC, 1/2 MIC, 1 MIC, and 2 MIC) were co-cultured for 0, 1, 2, 3, 4, and 5 h. The number of colonies was counted, and kinetics curves were constructed. (**B**): Transmission electron microscope and scanning electron microscopy analysis of APP. (**a**): Control untreated bacteria via TEM, the bar at the bottom right means 5.0 µm. High-magnification image of area indicated in upper right corner, the bar at the bottom right means 500 nm. (**b**): Bacteria treated with TP via TEM, the bar at the bottom right means 5.0 µm. High-magnification image of area indicated in upper right corner, the bar at the bottom right means 500 nm. (**c**): Untreated bacteria control analysis via SEM, the bar at the bottom right means 2.0 μm. (**d**): Bacteria treated with TP analysis via SEM, the bar at the bottom right means 2.0 μm. (**C**): The cell viability of NPTr cells of TP treatment. (**D**): The adherent ability of APP to NPTr cells or NPTr cells pretreatment with TP. Statistical analysis was performed by Student’s *t*-test. *n* = 3 in each group. Results are expressed as the mean ± SD of three independent experiments. * *p* < 0.05, *** *p* < 0.001.

**Figure 2 ijms-24-11842-f002:**
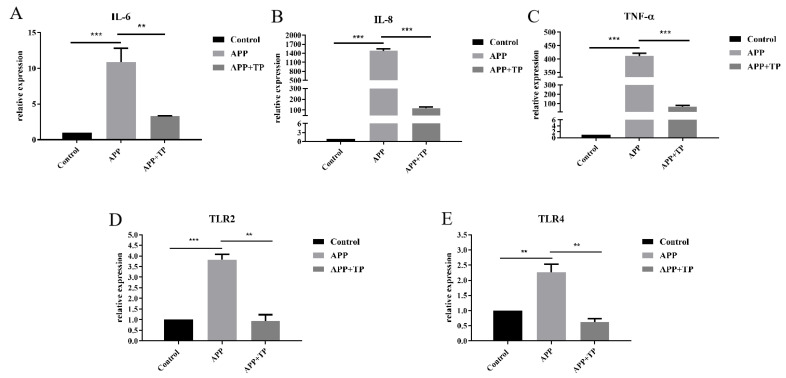
(**A**–**C**): Effect of APP on the Secretion of Inflammatory Factors in NPTr Cells or NPTr Cells Pretreated with TP. (**D**,**E**): The mRNA levels of TLR2 and TLR4 in NPTr cells and TP pretreatment cells infection with APP. Statistical analysis was performed by Student’s *t*-test. *n* = 3 in each group, ** *p* < 0.01, *** *p* < 0.001.

**Figure 3 ijms-24-11842-f003:**
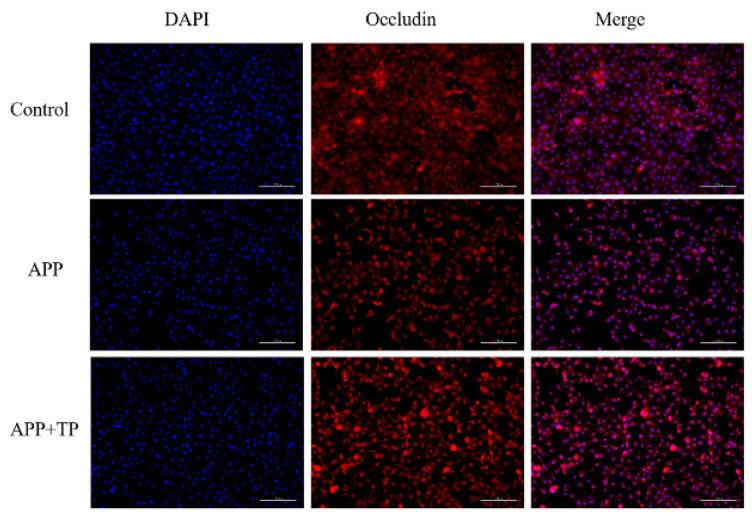
Immunofluorescence localization of occluding and β-catenin in NPTr cells. Scale bar indicated 100 μm.

**Figure 4 ijms-24-11842-f004:**
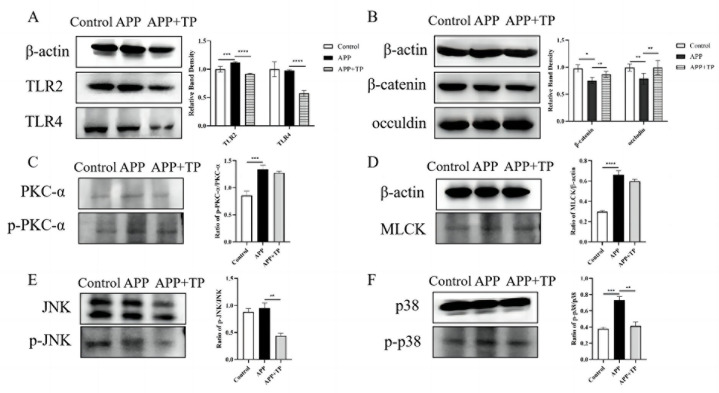
(**A**,**B**): The protein levels of TLR2, TLR4, ß-catenin, and occludin in NPTr cells and TP pretreatment cells infection with APP. GAPDH in whole cell lysates was detected as the loading control. (**C**): the ratio of p-PKC-α/PKC-α, (**D**): the ratio of MLCK: Effects of TP on PKC and MLCK pathways in NPTrs activated by APP. (**E**): the ratio of p-JNK/JNK, (**F**): the ratio of p-p38/p38: Effect of TP on phospho-JNK, and -p38 and total -JNK, and -p38 expression in NPTrs by Western blotting. (mean ± SD, *n* = 3).* *p* < 0.05, ** *p* < 0.01, *** *p* < 0.001, **** *p* < 0.0001.

**Figure 5 ijms-24-11842-f005:**
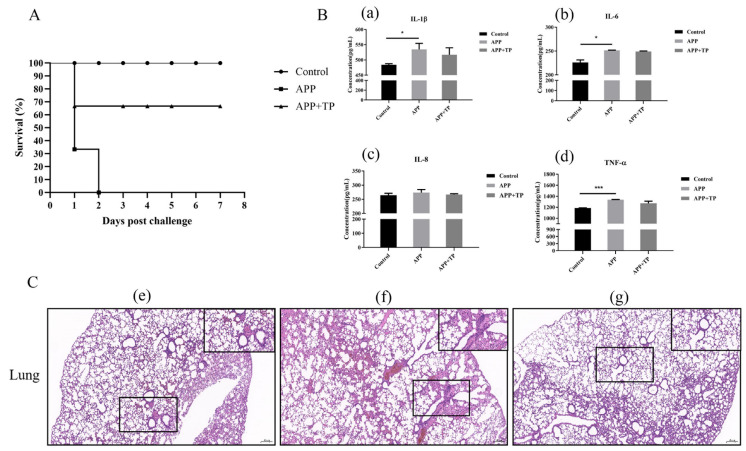
(**A**): Survival curves for mice in infection experiment. (**B**): Secretion of IL-1ß, IL-6, IL-8, and TNF-α in the serum of mice infected with APP, as measured by ELISA. * *p* < 0.05, *** *p* < 0.001. (**a**): IL-1β production in serum of APP-infected mice affected by TP. (**b**): IL-6 production in serum of APP-infected mice affected by TP. (**c**): IL-8 production in serum of APP-infected mice affected by TP. (**d**): TNF-α production in serum of APP-infected mice afected by TP. (**C**): Histopathology of representative lung tissues from BALB/c mice and TP pretreatment mice infected with APP. (**e**): control group. (**f**): mice with APP infection group. (**g**): mice administered with TP. The black band at the bottom left of each picture indicates the scale bar (200 μm).

**Figure 6 ijms-24-11842-f006:**
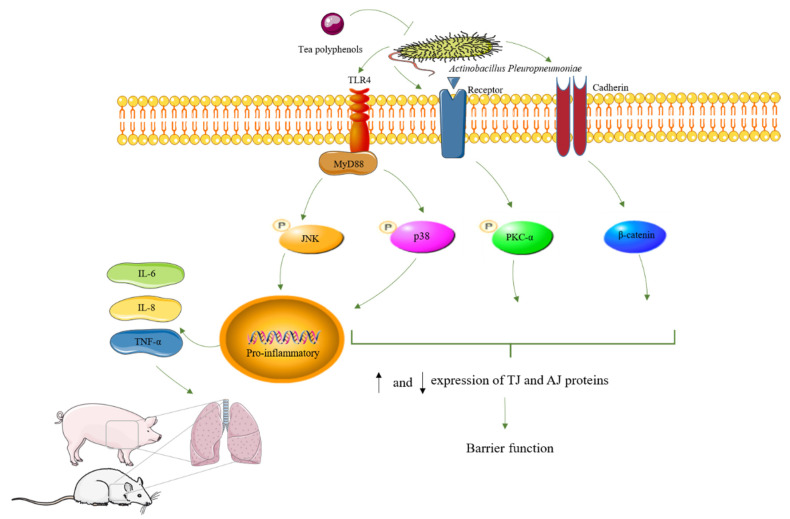
Schematic representation of TP in APP invasion of tracheal epithelial cells. “P” indicate phosphorylation. arrows (“↑” indicate increasing, “↓” indicate decreasing).

**Table 1 ijms-24-11842-t001:** Minimal inhibitory concentration (MIC) and minimal bactericidal concentration (MBC) values of TP for APP.

Medium	Compound	MIC (mg/mL)	MBC (mg/mL)
TSB	TP	0.625	1.25

**Table 2 ijms-24-11842-t002:** Primers used for qRT-PCR.

Gene	Nucleotide Sequence (5′-3′)	Tm (°C)
GAPDH	GGCTGCCCAGAACATCATCC	60
	GACGCCTGCTTCACCACCTTCTTG	
IL-6	GGAACGCCTGGAAGAAGATG	58
	ATCCACTCGTTCTGTGACTG	
IL-8	TTTCTGCAGCTCTCTGTGAGG	58
	CTGCTGTTGTTGTTGCTTCTC	
TNF-α	CGCATCGCCGTCTCCTACCA	60
	GACGCCTGCTTCACCACCTTCTTG	
TLR-2	ACGGACTGTGGTGCATGAAG	58
	GGACACGAAAGCGTCATAGC	
TLR-4	CATACAGAGCCGATGGTG	58
	CCTGCTGAGAAGGCGATA	

## Data Availability

The datasets generated and analyzed during the current study are available from the corresponding author on reasonable request.

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
