# Peer review of "Tea Polyphenols Protects Tracheal Epithelial Tight Junctions in Lung during Actinobacillus pleuropneumoniae Infection via Suppressing TLR-4/MAPK/PKC-MLCK Signaling"

_ijms, 2023, doi:10.3390/ijms241411842_

Round 1
Reviewer 1 Report
The manuscript entitled “Tea polyphenols protects tracheal epithelial tight junctions in lung during Actinobacillus Pleuropneumoniae infection via suppressing TLR-4/MAPK/PKC-MLCK signaling” by Xiaoyue Li et al, aimed to explore the therapeutic potential of tea polyphenol extract for the treatment of Actinobacillus pleuropneumoniae (APP) infections. In particular, they investigated the ability of TP to reduce inflammation, and pathogenicity of APP.
The work is not very well-written, but the results are interesting and contribute to explore the beneficial effects of tea polyphenols in animal health.
Comments:
In the Abstract section, please rephrase the following sentence “In this study, we investigated the effect of the anti-inflammatory, and antimicrobial, tea polyphenols (TP), …” because it is not well constructed. In the Line 23, “scanning electron microscope (SEM) assays, adherence assays, mouse assays” please rephrase to a less repetitive and well-written sentence.
Other similar errors are present in the main text.
I suggested to the author to characterize their extract in terms of polyphenol content and/or and amount of EGCG, EGC, ECG and EC by HPLC to improve the discussion. This information could be also useful to the scientific community to understand what the composition of TP is needed to reach the observed biological effect. Every extract is different and as a consequence the observed effects may vary.
The English need to be improved.
Author Response
Reviewer #1 (Comments for the Author):
The manuscript entitled “Tea polyphenols protects tracheal epithelial tight junctions in lung during Actinobacillus Pleuropneumoniae infection via suppressing TLR-4/MAPK/PKC-MLCK signaling” by Xiaoyue Li et al, aimed to explore the therapeutic potential of tea polyphenol extract for the treatment of Actinobacillus pleuropneumoniae (APP) infections. In particular, they investigated the ability of TP to reduce inflammation, and pathogenicity of APP.
The work is not very well-written, but the results are interesting and contribute to explore the beneficial effects of tea polyphenols in animal health.
In the Abstract section, please rephrase the following sentence “In this study, we investigated the effect of the anti-inflammatory, and antimicrobial, tea polyphenols (TP), …” because it is not well constructed.
RE: Thank you for your nice suggestion. We have rephrased the following sentence “In this study, we investigated the effect of the anti-inflammatory, and antimicrobial, tea polyphenols (TP), …” into “In this study, we investigated the effect of tea polyphenols (TP) against APP .” in the revised manuscript.
In the Line 23, “scanning electron microscope (SEM) assays, adherence assays, mouse assays” please rephrase to a less repetitive and well-written sentence.
RE: Thank you very much for your advice. We have rephrased the sentence “scanning electron microscope (SEM) assays, adherence assays, mouse assays” into “MIC and MBC of TP showed significant inhibitory effects on bacteria growth and cellular damage of APP . Furthermore, TP decreased not only the adherent activity of APP to the newborn pig tracheal epithelial cells Nptr but also the destruction to the tight adherence junction proteins β-catenin and occludin. Moreover, TP improved the survival rate of APP infected mice and attenuated the release of the inflammation‑related cytokines IL-6, IL-8, and TNF-α.”in the revised manuscript.
Other similar errors are present in the main text.
RE: Thank you for your kind suggestion. We have corrected the main text in the revised manuscript.
I suggested to the author to characterize their extract in terms of polyphenol content and/or and amount of EGCG, EGC, ECG and EC by HPLC to improve the discussion. This information could be also useful to the scientific community to understand what the composition of TP is needed to reach the observed biological effect. Every extract is different and as a consequence the observed effects may vary.
RE: Thank you for your good suggestion. We have added information about the extract of tea polyphenols content and/or and the amount of EGCG, EGC, ECG and EC by HPLC in the revised manuscript.
Reviewer 2 Report
The paper is an interesting work which investigates the protective activity antipsychotic activity of TP against AP infection by negatively modulating the TLR signalling involving different cytoplasmatic kinases.
TP were evaluated in several biochemical assays and in vivo that confirmed the quality of the proposed study.
The paper is well written and organized. References are adequate.
The paper needs to be improved before its acceptance, as follows:
- In the introduction, the Authors should describe the antibiotic drugs (drugs classes) against which APP strains have developed resistance.
- Make a look into the chemical composition of TP: what are the main components to which the Authors could attribute the observed protective properties?
TP from what Tea? Provide more details!
Please refer to this manuscript Shiming Li, Liang Zhang, Xiaochun Wan, Jianfeng Zhan, Chi-Tang Ho. Focusing on the recent progress of tea polyphenol chemistry and perspectives. Food Science and Human Wellness 2022, 11, 437-444, https://doi.org/10.1016/j.fshw.2021.12.033
- Regarding the ability of TP to inhibit APP growth in vitro, the Authors did not include a reference compound (antibiotic), also for a comparative analysis.
- In the evaluation of cell viability in Figure 1C, the Authors used a 100-fold lower concentrations of TP (0.00625 mg/mL, 0.125 mg/mL) than those corresponding to MIC and MBC (0.625 mg/mL and 1.25 mg/mL). Why?
Author Response
Reviewer 2
The paper is an interesting work which investigates the protective activity antipsychotic activity of TP against AP infection by negatively modulating the TLR signalling involving different cytoplasmatic kinases.
TP were evaluated in several biochemical assays and in vivo that confirmed the quality of the proposed study.
The paper is well written and organized. References are adequate.
The paper needs to be improved before its acceptance, as follows:
- In the introduction, the Authors should describe the antibiotic drugs (drugs classes) against which APP strains have developed resistance.
RE: Thank you for your nice suggestion. We have added the content about the antibiotic drugs (drugs classes) against APP strains have developed resistance in the introduction of revised manuscript.
- Make a look into the chemical composition of TP: what are the main components to which the Authors could attribute the observed protective properties?
RE: Thank you for your nice suggestion. We considered the EGCG, EGC, ECG and EC are the main components that attributed the observed protective properties.
- TP from what Tea? Provide more details!
RE: Thank you for your nice suggestion. In this study, TP extracted from green tea, we have added information about their extract in terms of tea polyphenols content in the revised manuscript.
Please refer to this manuscript Shiming Li, Liang Zhang, Xiaochun Wan, Jianfeng Zhan, Chi-Tang Ho. Focusing on the recent progress of tea polyphenol chemistry and perspectives. Food Science and Human Wellness 2022, 11, 437-444, https://doi.org/10.1016/j.fshw.2021.12.033
RE: Thank you for your nice suggestion. We have refered and quoted to this manuscript Shiming Li, Liang Zhang, Xiaochun Wan, Jianfeng Zhan, Chi-Tang Ho. Focusing on the recent progress of tea polyphenol chemistry and perspectives. Food Science and Human Wellness 2022, 11, 437-444, https://doi.org/10.1016/j.fshw.2021.12.033
Regarding the ability of TP to inhibit APP growth in vitro, the Authors did not include a reference compound (antibiotic), also for a comparative analysis.
RE: Thank you for your nice suggestion. We added to compounds (antibiotics) for comparative analysis in the revised manuscript.
In the evaluation of cell viability in Figure 1C, the Authors used a 100-fold lower concentrations of TP (0.00625 mg/mL, 0.125 mg/mL) than those corresponding to MIC and MBC (0.625 mg/mL and 1.25 mg/mL). Why?
RE: Thank you for your nice suggestion. Taking into account the effects of of higher concentrations of TP on cellular activity and the cost of the experiment, we used a 10-fold ,not 100-fold lower concentrations of TP (Figure 1C).

Round 2
Reviewer 1 Report
The authors made all required changes. In my opinion, the manuscript is now suitable for the publication.
The English language required minor only minor editing.
Reviewer 2 Report
The paper was properly improved by the Authors, so I suggest its publication in its current form